# Computational modeling and experimental validation of the EPI-X4/CXCR4 complex allows rational design of small peptide antagonists

Pandian Sokkar [1,2,8], Mirja Harms[3,8], Christina Stürzel[3], Andrea Gilg[3], Gönül Kizilsavas[4], Martina Raasholm[5], Nico Preising[6], Manfred Wagner[4], Frank Kirchhoff[3], Ludger Ständker [6], Gilbert Weidinger [5], Benjamin Mayer[7], Jan Münch [3,6✉] & Elsa Sanchez-Garcia [1✉]

EPI-X4, a 16-mer fragment of albumin, is a specific endogenous antagonist and inverse agonist of the CXC-motif-chemokine receptor 4 (CXCR4) and thus a key regulator of CXCR4 function. Accordingly, activity-optimized synthetic derivatives of EPI-X4 are promising leads for the therapy of CXCR4-linked disorders such as cancer or inflammatory diseases. We investigated the binding of EPI-X4 to CXCR4, which so far remained unclear, by means of biomolecular simulations combined with experimental mutagenesis and activity studies. We found that EPI-X4 interacts through its N-terminal residues with CXCR4 and identified its key interaction motifs, explaining receptor antagonization. Using this model, we developed shortened EPI-X4 derivatives (7-mers) with optimized receptor antagonizing properties as new leads for the development of CXCR4 inhibitors. Our work reveals the molecular details and mechanism by which the first endogenous peptide antagonist of CXCR4 interacts with its receptor and provides a foundation for the rational design of improved EPI-X4 derivatives.

[1] Computational Biochemistry, Center of Medical Biotechnology, University of Duisburg-Essen, Essen, Germany. [2] Faculty of Allied Health Sciences, Chettinad Hospital and Research Institute, Chettinad Academy of Research and Education, Kelambakkam, Tamil Nadu 603103, India. [3] Institute of Molecular Virology, Ulm University Medical Center, Ulm, Germany. [4] Max Planck Institute for Polymer Research, Mainz, Germany. [5] Institute of Biochemistry and Molecular Biology, Ulm University, Ulm 89081, Germany. [6] Core Facility Functional Peptidomics, Ulm University Medical Center, Ulm 89081, Germany. [7] Institute for Epidemiology and Medical Biometry, Ulm University, Ulm 89075, Germany. [8] These authors contributed equally: Pandian Sokkar, Mirja Harms. ✉email: Jan.Muench@uni-ulm.de; elsa.sanchez-garcia@uni-due.de

Chemokine receptors are important mediators of numerous processes in the human body. Among them is C-X-C motif chemokine receptor 4 (CXCR4), a 365-residue rhodopsin-like G-protein coupled receptor (GPCR). CXCR4 is expressed on ubiquitous hematopoietic and non-hematopoietic tissues where it regulates important processes, such as immune response, development, hematopoiesis, vascularization, tissue renewal, and regeneration[1]. It is therefore not surprising that the faulty regulation of CXCR4 is responsible for several pathologies, i.e., inflammatory diseases[2], immunodeficiencies, or cancer[3,4]. In many different forms of cancer, CXCR4 is often overexpressed or overactivated[1,5], which is linked to cancer progression by promoting proliferation, survival, and metastasis[6,7]. CXCR4-expressing cells migrate in direction of CXCL12, the sole endogenous chemokine ligand of CXCR4. CXCL12 is mainly expressed in the bone marrow, the lymph nodes, lung, and liver; tissues where primary metastasis mainly occur[8]. Besides that, CXCR4 is a major entry coreceptor of HIV-1[9]. Thus, CXCR4 represents a promising drug target. In the last years, intense research was performed to identify and develop CXCR4 antagonists for therapeutic applications in HIV/AIDS, cancer, and inflammatory disorders. However, the only FDA-approved CXCR4 antagonist so far is AMD3100, which is restricted to single treatments due to its severe side effects[10].

Because of the importance of CXCR4 in physiological and pathological processes, crystal structures of CXCR4 in complex with different ligands have been determined; (i) with the viral chemokine antagonist vMIP-II[11], (ii) with the small molecule antagonist IT1t[12], and (iii) with the cyclic peptide CVX15 (analog of polyphemusin)[12]. As a member of the GPCR family, the structure of CXCR4 consists of a canonical bundle of seven transmembrane (TM) α-helices, three intracellular (ICL), and three extracellular (ECL) loops. The ECL N-terminus of CXCR4 features a 34-residues intrinsically disordered loop that forms a disulfide bridge with C274 of helix VII through C28. CXCR4 has a relatively large and open binding pocket that is located at the ECL region[8,12]. This binding pocket is defined by the seven TM domains and is decorated by negatively charged aspartate and glutamate residues[11]. It can be separated into loosely defined major and minor subpockets; the first comprised of TMs III, IV, V, VI, and VII and the latter comprised of TMs I, II, III, and VII[13]. While in GPCRs most ligands only interact with the major subpocket, for CXCR4 it was shown that the small molecule antagonist IT1t[12], as well as the chemokines vMIP-II and CXCL12[11], interact with the minor subpocket. In the case of CXCL12, the N-terminus of CXCR4 mediates binding and is responsible for receptor activation[11,14]. $K1_{CXCL12}$ interacts with $D97_{CXCR4}$ and $E288_{CXCR4}$ through the N-terminal amine group of CXCL12. Another residue that is thought to mediate receptor activation is $D187_{CXCR4}$ (ECL2) that interacts with $S4_{CXCL12}$ and $Y7_{CXCL12}$[14].

Zirafi et al. identified an endogenous antagonist of CXCR4 termed EPI-X4 (endogenous peptide inhibitor of CXCR4) by screening a peptide library derived from human hemofiltrate for inhibitors of CXCR4-tropic HIV-1 infection[15,16]. They found that this 16-mer peptide is derived by proteolytic degradation of human serum albumin by pH-regulated proteases, e.g., cathepsin D, and is therefore endogenously present at acidic sites of the body. EPI-X4 binds specifically to CXCR4 and interrupts the interaction of CXCL12 with its receptor thereby antagonizing chemokine-mediated effects, like cell migration and infiltration. Additionally, EPI-X4 has inverse antagonistic properties, since it reduces basal receptor signaling activity. So far, the physiological function of EPI-X4 is not clear, however, the peptide was found in the urine of patients with renal failure and therefore might have regulatory functions in the body[16].

EPI-X4 represents a promising candidate for development as a therapeutic CXCR4 antagonist: Compared to the only licensed CXCR4 inhibitor Plerixafor (AMD3100)[17], which also binds CXCR7 and is associated with adverse events if applied over prolonged periods of time[18], endogenous EPI-X4 exclusively targets CXCR4 and is not associated with toxicity[16]. Moreover, EPI-X4 also acts as an inverse agonist of CXCR4 which may be particularly relevant for the therapy of CXCR4-linked disorders caused by receptor activating mutations such as in Waldenström's macroglobulinemia[16,19,20].

Accordingly, first and second-generation EPI-X4 derivatives were developed and evaluated in animal models of CXCR4-linked diseases. First-generation analogs of EPI-X4 include the C-terminally truncated 12-mer derivative WSC02 that harbors four amino acid substitutions compared to the wild type (L1I, Y4W, T5S, and Q10C)[16]. EPI-X4 WSC02 has about 30-fold increased potency compared to the wild-type peptide[16]. More recently, we developed an optimized 12-mer version, EPI-X4 JM#21. EPI-X4 JM#21 harbors three additional amino acid substitutions (V2L, K6R, and V9L), which led to a further increase in receptor affinity with respect to EPI-X4[20]. In addition, EPI-X4 JM#21 showed increased efficiency for inhibition of CXCL12-mediated receptor signaling (>100-fold compared to EPI-X4, tenfold compared to WSC02) and cancer cell migration (~1500-fold compared to EPI-X4 and 30-fold compared to WSC02). Optimized EPI-X4 derivatives also showed promising therapeutic effects in mouse models of stem cell mobilization, allergic airway eosinophilia, and atopic dermatitis[16,20].

Computational modeling has been carried out to determine the interaction sites of different CXCR4 receptor ligands with the binding pockets of CXCR4[21,22]. Several molecular dynamics (MD) studies are reported, primarily (i) to characterize the structure and function of CXCR4 and agonist binding[23–25], (ii) to study the interactions between CXCR4 and small molecule/peptide antagonists[26–28], and (iii) to address the dimerization of CXCR4[29–31]. However, the interactions of EPI-X4 with CXCR4 have so far not been investigated in detail. Recently, we reported optimized derivatives of EPI-X4, based on a model of CXCR4/EPI-X4 obtained by molecular docking studies[20]. While this preliminary docking model was able to successfully explain the role in the binding of key EPI-X4 residues such as K6 and K7, it did not provide information on the role of different regions of the peptide and other interaction motifs. Further, key factors such as the dynamics of the peptide–protein complex, membrane and solvent effects were not considered by the docking model. We also note that a 16-amino acids linear peptide such as EPI-X4 has a substantial conformational landscape, consisting of many local minima. Therefore, a setback of merely using docking approaches is that the complex would be getting trapped in a local minimum. For a detailed understanding of how EPI-X4 and its derivatives bind and block CXCR4, a thorough computational study is imperative.

To this end, we carried out coarse-grained (CG) studies and extensive all-atom MD simulations of CXCR4/EPI-X4 models in an explicit membrane-water environment. The simulations, which allowed to identify the interaction motifs of the endogenous peptide EPI-X4 with CXCR4, were complemented with site-directed mutagenesis and activity studies to better understand EPI-X4's function and effects on the receptor. We also report the design of shorter and highly efficient EPI-X4 derivatives based on the knowledge provided by the biomolecular simulations. The design of shorter active peptides is a key step towards biomedical applications, in terms of synthetic accessibility, costs, dosage, and optimization for oral delivery.

## Results

**Binding modes of EPI-X4 to CXCR4**. We used various computational approaches to determine the binding mode of EPI-X4 with CXCR4, explicitly considering solvation and the membrane environment (Fig. 1a). First, we performed docking calculations with CXCR4 (PDB IDs: 3ODU and 2K04, see computational details)[12,32] using three different conformations of EPI-X4 taken from the previously published NMR ensemble (2N0X)[16]. We generated another binding mode by homology modeling, using the crystal structure of the vMIP-II peptide/CXCR4 complex (4RWS)[11] as a template. For ease of discussion, these four binding poses are abbreviated as MID-IN1, MID-IN2, CTER-IN, and NTER-IN (Fig. 1b–e). Although there are many conformations possible for this system, these models provided a comprehensive starting point for MD simulations to assess the interactions between CXCR4 and EPI-X4.

**All-atom MD simulations**. We performed all-atom MD simulations of the four systems (MID-IN1, MID-IN2, CTER-IN, and NTER-IN, Fig. 1), three replicas in each case. We used full atom resolution, including explicit solvent and the lipid bilayer (Fig. 1a). The analysis of the MDs indicated that the CTER-IN and NTER-IN motifs showed the lowest amount of structural variations in most of the trajectories, also suggesting that these two binding modes are favorable (Fig. S1 and Supplementary computational details). Therefore, we extended the cumulative simulation time for CTER-IN and NTER-IN from 600 ns to 1.35 μs each. The NTER-IN binding mode showed negligible fluctuation in residues 1–8 (Fig. S1c), which could be critical for the binding of EPI-X4 derivatives[16].

The solvent accessible surface area (SASA) of the peptide during the MD simulations (Table S1 and Fig. S2) indicates more solvent exposure of the peptide in the cases of MID-IN1 and MID-IN2, whereas in the CTER-IN and NTER-IN modes, EPI-X4 is buried slightly deeper into the receptor (Table S1). In addition, the larger protein–peptide interaction interface in the CTER-IN and NTER-IN motifs suggests that the interaction of EPI-X4 with CXCR4 is stronger in these cases with respect to MID-IN1 and MID-IN2.

Next, we carried out an RMSD-based clustering analysis of the simulation trajectories (Fig. 2). The MID-IN2 motif did not produce any meaningfully populated cluster (5% for the highest populated cluster). MID-IN1 resulted in a cluster of conformations 22% populated in ~600 ns simulation (Fig. 2a). CTER-IN exhibited two different clusters, both of which are significantly populated (32 and 31%) in 1.35 μs. The difference between these two clusters is that in cluster#1 (Fig. 2b) EPI-X4 interacts in the "major binding pocket" of CXCR4, whereas in cluster#2 (Fig. 2c) EPI-X4 partially occupies both the "minor" and "major" binding pockets of the receptor. Notably, in the case of the NTER-IN mode, which shows the highest population of the conformations (70%) in a single cluster in 1.35 μs, the N-terminal region of EPI-X4 interacts with the "minor pocket" of CXCR4 (Fig. 2d), in a similar manner to the vMIP-II peptide[11]. We note that the same tendencies were observed if only 600 ns trajectories of CTER-IN and NTER-IN were analyzed (i.e., CTER-IN: 31% and NTER-IN: 86%). Thus, our results indicate that although it may visit the major pocket, EPI-X4 primarily targets the minor binding pocket of CXCR4.

Next, we focused on the specific interactions that stabilize the complexes of EPI-X4 and CXCR4. We calculated the number of hydrogen bonds (averaged over the simulation time) formed between CXCR4 and EPI-X4 during the MD simulations. The NTER-IN binding motif exhibited the highest number of hydrogen bonding interactions with an average of five hydrogen bonds. MID-IN1 and MID-IN2 featured a smaller number of hyrogen-bond contacts (average of 2.6 and 3.1 hydrogen bonds, respectively). By comparison, CTER-IN shows a slightly higher number of hydrogen bonds (average 3.8). This result agrees with the previous findings (RMSD, SASA, peptide–protein surface, and clustering analysis) that indicate that MID-IN and MID-IN2 are not favored binding modes when compared to the CTER-IN and NTER-IN complexes.

As mentioned, NTER-IN shows the highest population of hydrogen-bonding contacts (Table S2). The H bond involving T5$_{EPI-X4}$ and D187$_{CXCR4}$ sidechains was found to be populated by 69.5% during the 1.35 μs simulation of the NTER-IN mode. Similarly, hydrogen bonds involving L1 (mainchain), R3, T5, K6, and K7 residues of EPI-X4 were also found to be significantly populated in this mode. This suggests that these residues are playing a pivotal role in the interaction with the predominantly negatively charged binding pocket of CXCR4. These findings are in agreement with mutagenesis experiments indicating that L1 and K6 are key for the activity of the peptide[16,20]

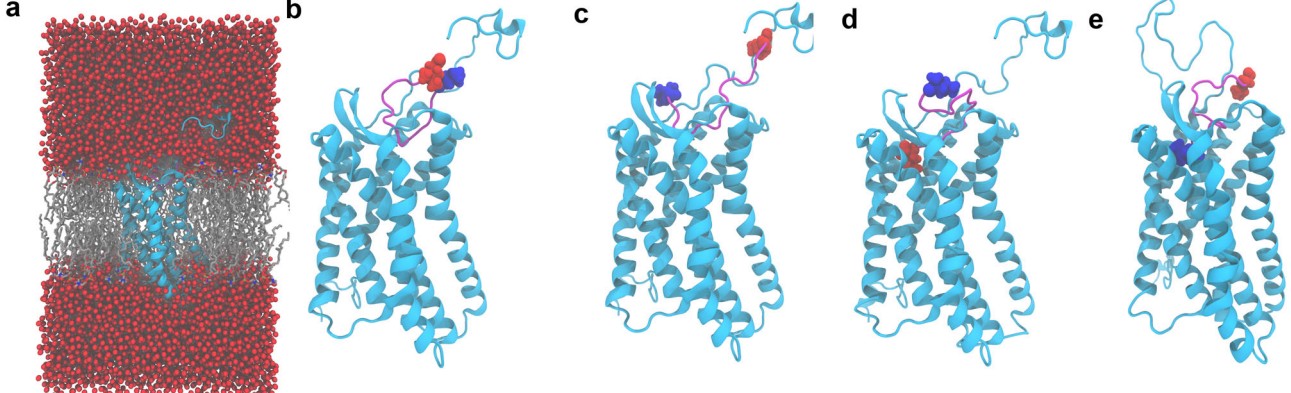

**Fig. 1 MD simulations setup and binding modes of the CXCR4/EPI-X4 complex.** CXCR4 (cyan) and EPI-X4 (purple) are shown in cartoon representation and the N-terminal (blue) and C-terminal (red) residues of EPI-X4 are shown as spheres. **a** Representative all-atom CXCR4/EPI-X4 complex embedded in a POPC (1-palmitoyl-2-oleoyl-sn-glycero-3-phosphocholine) lipid bilayer (gray sticks) and water (red spheres) as used in the MD simulations. Bilayer and water are omitted in the other figures of this manuscript for clarity. The initial binding modes represented in (**b-d**) were obtained using docking calculations and the binding mode shown in (**e**) was obtained using homology modeling. In (**b**, **c**) the middle portion of the peptide is inside the binding pocket (abbreviated as MID-IN1 and MID-IN2, respectively. In **d** the C-terminus of the peptide is inside the binding pocket (CTER-IN) while in (**e**), the N-terminus of EPI-X4 is the region inside the binding pocket (NTER-IN).

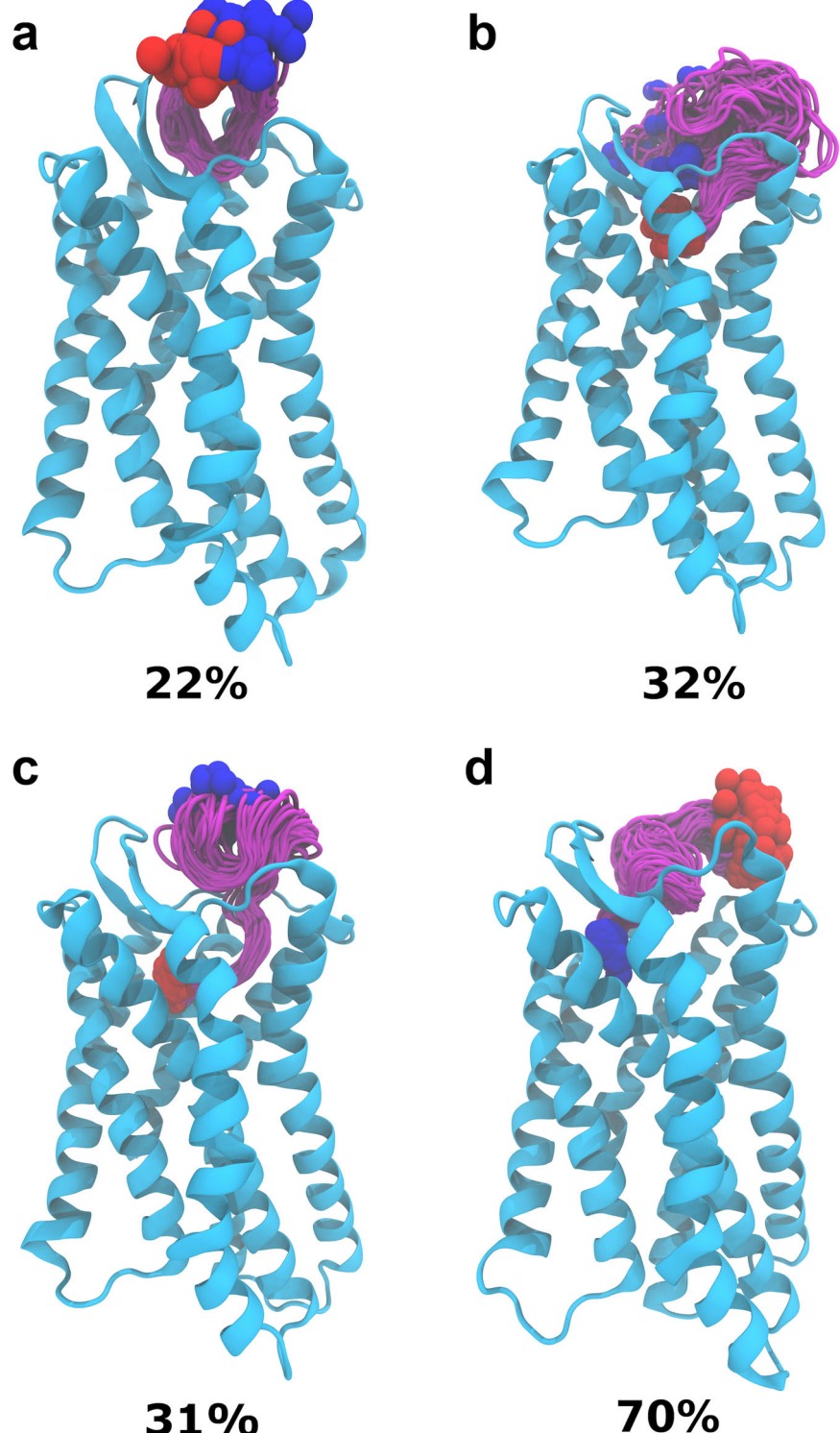

**Fig. 2 Clustering analysis of peptide conformations in MD trajectories.** Highest populated clusters of MID-IN1 (**a**), CTER-IN (**b** and **c**) and NTER-IN (**d**) are shown. The color scheme is the same as in Fig. 1. Structures of the same cluster are shown to illustrate the spread of conformations. The clustering analysis was performed with an RMSD cutoff of 3 using the VMD program[48].

In addition, the analysis of the total interaction energies and the corresponding van der Waals (vdW) and electrostatic contributions indicates that NTER-IN exhibits high interaction strength compared to the other binding modes, followed by CTER-IN (Table S3). Interestingly, the vdW contribution to the interaction energy is the same in NTER-IN and CTER-IN. Thus, the difference in their interaction strength is mainly due to electrostatic contributions. This trend is in good agreement with the previously discussed results, all of them pointing to NTER-IN as the most favorable binding mode.

Next, we decomposed the interaction energies by residue-wise contributions (Fig. S3). Our analysis indicates that positively charged residues provide the largest favorable contribution to the interaction energies. This is because the binding pocket of CXCR4

is rich in negatively charged residues. Thus, the positively charged N-terminal Leu (L1) of EPI-X4 stabilizes the binding, whereas the negatively charged C-terminal Leu (L16) has a destabilizing effect. The contributions to the interaction energy of the positively charged residues L1, R3, K6, and K7 of EPI-X4 are nearly the same in all binding modes. Consequently, the improved binding affinity of the NTER-IN interaction motif may be related to the optimal positioning of the N-terminal residues V2, Y4, and T5.

**EPI-X4/CXCR4 interaction sites**. Zirafi et al.[16] reported that mutations of L1 (L1A, L1G, and L1F) or deletion of this amino acid render EPI-X4 inactive. On the contrary, truncations at the C-terminus of up to four residues did not seem to influence CXCR4 binding or inhibition. It was also shown that the deletion of C-terminal residues Q10–L16 did not affect the binding drastically. Analysis of our structural models (Fig. 3a–d) shows that the sidechain of L1 in NTER-IN mode fits effortlessly in the hydrophobic minor pocket area (Fig. 3c, d), which allows rationalizing the lack of activity of the aforementioned N-terminal mutations. With respect to EPI-X4, the L1F derivative has a larger and planar phenyl ring that would be sterically hindered in the hydrophobic pocket of CXCR4. On the contrary, L1A and L1G, that render the peptide inactive, have very small sidechains that cannot establish optimal contacts in such pocket, explaining the lack of activity.

Unlike NTER-IN, the binding modes MID-IN, MID-IN2, and CTER-IN do not agree with the experimental findings stated above. This is because, in these three models, the N-terminal region of the peptide is exposed to the solvent (Figs. 1, 3). However, L1 was found to contribute to the stabilization of the complex in all binding modes (Fig. S3). We decomposed the L1 interaction energy to identify the sidechain contribution, which was found to be 0.0, −0.7, and −1.3 kcal/mol, for MID-IN, MID-IN2, and CTER-IN, respectively. For NTER-IN, the hydrophobic L1 sidechain contributes −6.5 kcal/mol to the interaction energy, indicating that the L1 is optimally placed in terms of both electrostatic and hydrophobic interactions.

In NTER-IN, the sidechain of $L1_{EPI-X4}$ is buried in the hydrophobic cavity of the "minor" binding pocket of CXCR4 (formed by F93, W94, W102, V112, and Y116), while the backbone ammonium group of EPI-X4 is forming a salt bridge with D97 of CXCR4 (Fig. 3d). This arrangement also allows the peptide to establish a variety of interactions, such as the $R3_{EPI-X4}$–$H281_{CXCR4}$ hydrogen bond, the salt bridges $K6_{EPI-X4}$–$D187_{CXCR4}$, $K7_{EPI-X4}$–$D262_{CXCR4}$, and $L16(C-terminal)_{EPI-X4}$–$K271_{CXCR4}$ (Table S2). Another interesting aspect of this binding mode is the additional stabilization provided by the short β-strand involving residues V11–T15 of EPI-X4, which interacts with the β-strand of CXCR4 comprising residues E25–R30 (Fig. 3c). These interactions were either absent or non-conserved in the other binding modes.

In addition to the all-atom studies, we performed extensive CG MD simulations with biasing forces to allow for the self-assembly of EPI-X4 into the binding pocket of CXCR4 (Supplementary Methods). By far, most of the productive trajectories lead to the NTER-IN binding mode, which was easily accessible by the

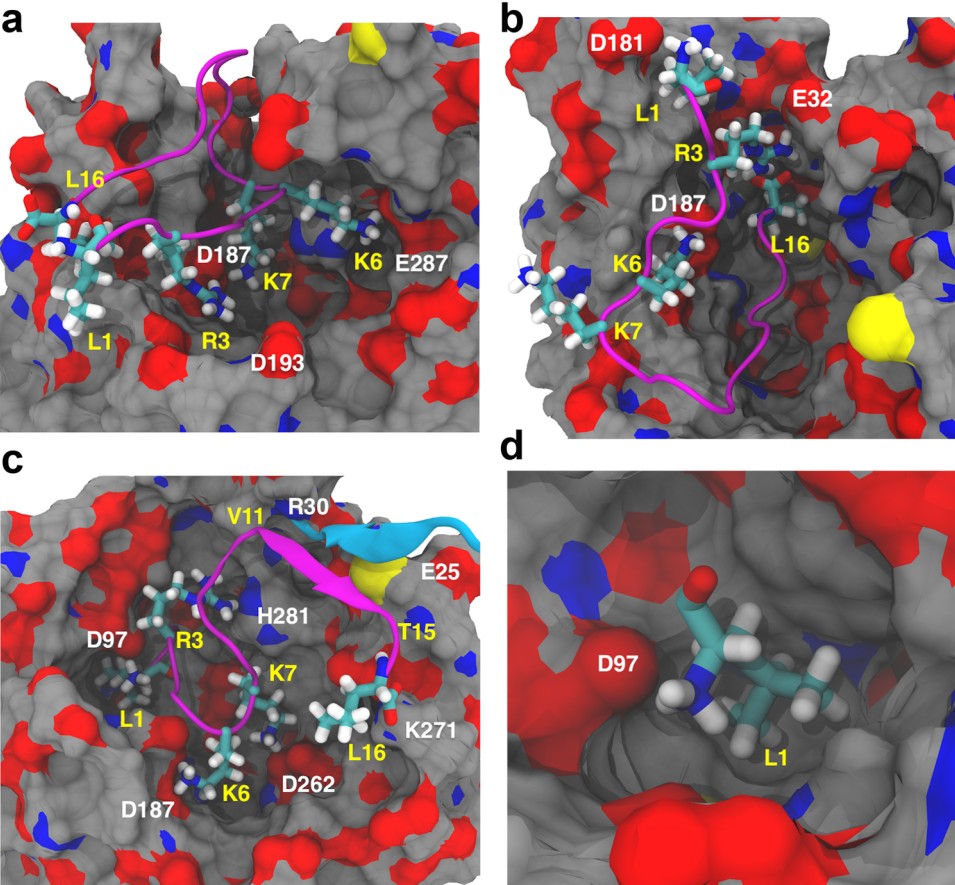

**Fig. 3 Analysis of the interactions between CXCR4 and EPI-X4 in the MD simulations.** Binding modes MID-IN1 (**a**), CTER-IN (**b**), and NTER-IN (**c**, **d**) were identified from the clustering analysis. CXCR4 is represented as surface (carbon = gray, oxygen = red, and nitrogen = blue) and EPI-X4 is shown in cartoon diagram (purple). Important residues of EPI-X4 are shown as sticks and labeled in yellow. CXCR4 residues involved in the binding are labeled in white. The β-strand of CXCR4 comprising residues E25–R30 is shown in cyan.

peptide (Fig. S4). Thus, the CG studies further established NTER-IN as the most favorable binding mode of EPI-X4 in CXCR4.

**Site-directed mutagenesis experiments confirm the results of biomolecular simulations**. To further assess these findings, we performed site-directed mutagenesis experiments of the CXCR4 residues predicted to be involved in the interaction with EPI-X4 (Fig. 4).

Mutated CXCR4 constructs were transfected into 293 T cells that have a very low endogenous expression level of wild-type CXCR4. Transfected cells were then analyzed for CXCR4 surface expression (Fig. S5a) and EPI-X4 binding using a flow cytometry-based antibody competition assay (Fig. S5b, c)[33]. This assay is based on the competition of the monoclonal CXCR4 antibody 12G5 (an antibody that binds to a region close to the receptor orthosteric binding pocket) and CXCR4 ligands for receptor interaction[33]. Mutations of residues predicted to be important for EPI-X4 binding strongly decreased receptor binding affinity (increase of 50% inhibitory concentration, $IC_{50}$, Fig. 4a, b and Table 1). When $D97_{CXCR4}$ was replaced by uncharged residues (D97S, D97T, D97N, and D97Q), the binding was nearly abolished, indicating that this residue is essential for the binding of EPI-X4. As predicted, the D97E mutation causes a reduction of the binding affinity of EPI-X4 to CXCR4 (~20-fold), despite the high similarity between the amino acids Asp and Glu. This can be

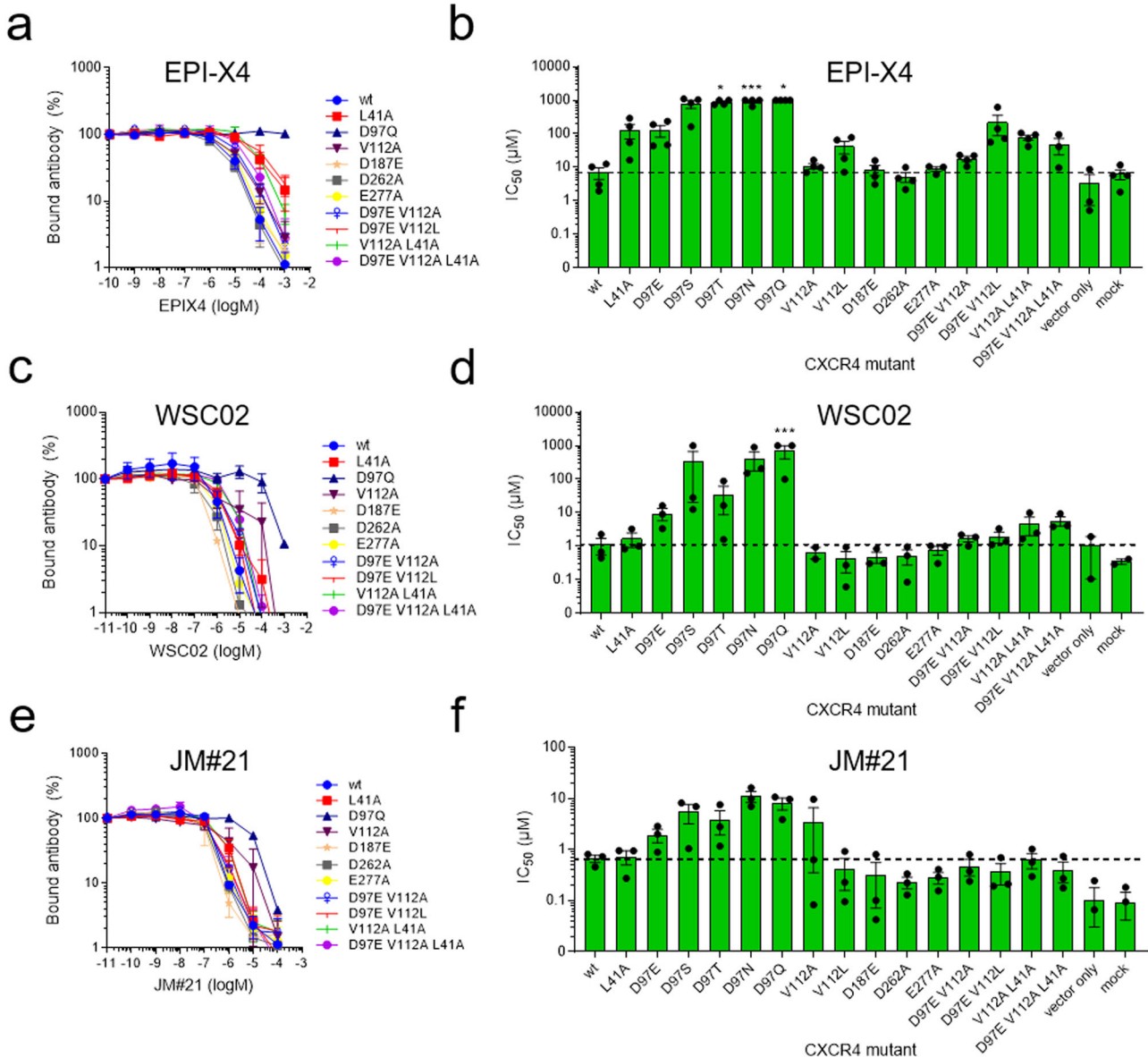

**Fig. 4 The interactions of EPI-X4 and the EPI-X4 derivatives WSC02 and JM#21 with point-mutated CXCR4 were assessed by an antibody competition assay.** Amino acid substitutions were introduced in the sequence of CXCR4 by site-directed mutagenesis, cloned into an IRES-eGFP expression vector, and transfected into 293 T cells. Afterward, cells were incubated with serially diluted EPI-X4 (**a**, **b**), WSC02 (**c**, **d**), or JM#21 (**e**, **f**) in presence of a constant concentration of CXCR4 specific antibody (clone 12G5). After 2 h, bound antibody was analyzed by flow cytometry. **a**, **c**, **e**) Representative dose-dependent replacement of 12G5 antibody by peptides. **b**, **d**, **f**) $IC_{50}$ values were calculated by nonlinear regression. Shown are data derived from at least three individual experiments ± SEM. $IC_{50}$ values were set to 1000 μM (EPI-X4 and WSC02) or 100 μM (JM#21) if the value exceeds the maximally used concentration. *$p \leq 0.05$, ***$p \leq 0.001$ (one-way ANOVA for comparison with wt). For $IC_{50}$ values see Table 1. For all binding curves see also Fig. S8.

**Table 1 IC$_{50}$ values of EPI-X4 determined in a 12G5-competition assay for CXCR4 mutants.**

| CXCR4 mutation | IC$_{50}$ ± SEM (µM) | Fold change | Hill slope[a] | Interaction with EPI-X4[b] | Agreement[c] |
|---|---|---|---|---|---|
| wt | 6.99 ± 2.63 | | −1.67 ± 0.50 | - | |
| L41A | 131.83 ± 61.64 | 19 | −1.02 ± 0.16 | V2 sidechain, hydrophobic | yes |
| D97E | 129.08 ± 49.34 | 18 | −0.96 ± 0.11 | L1 N-terminal, salt bridge | yes |
| D97S | 784.78 ± 214.89 | 112 | Nd | L1 N-terminal, salt bridge | yes |
| D97T | >1000 | - | Nd | L1 N-terminal, salt bridge | yes |
| D97N | >1000 | - | Nd | L1 N-terminal, salt bridge | yes |
| D97Q | >1000 | - | Nd | L1 N-terminal, salt bridge | yes |
| V112A | 10.96 ± 1.98 | 1.6 | Nd | L1 sidechain, hydrophobic | yes |
| V112L | 41.12 ± 17.33 | 6 | −1.03 ± 0.14 | L1 sidechain, hydrophobic | yes |
| D187E | 8.52 ± 2.84 | 1.2 | −1.25 ± 0.13 | T5 sidechain, H bonds | yes |
| D262A | 5.21 ± 1.67 | 0.7 | −1.05 ± 0.04 | K7 sidechain, salt bridge | no |
| E277A | 8.91 ± 1.43 | 1.3 | −1.21 ± 0.28 | T5 sidechain, H bonds | yes |
| D97E + V112A | 18.01 ± 2.95 | 2.6 | −1.20 ± 0.15 | - | yes |
| D97E + V112L | 222.74 ± 133.83 | 32 | −1.07 ± 0.15 | - | yes |
| V112A + L41A | 78.04 ± 13.91 | 11 | −1.28 ± 0.26 | | yes |
| D97E + V112A + L41A | 48.44 ± 24.73 | 7 | −0.93 ± 0.16 | | yes |

[a]Hill slopes were determined in GraphPad Prism using nonlinear regression curve fit. $P > 0.05$ for all hill slopes compared to wt (one-way ANOVA with Dunnett's post hoc test).
[b]EPI-X4 residues interacting with wild-type residues of CXCR4 in the NTER-IN model.
[c]Agreement between the NTER-IN model and the mutagenesis experiments.

related to the slightly extended (by one –CH$_2$– group) sidechain of Glu, which weakens the salt bridge with L1$_{EPI-X4}$ and/or the hydrophobic interaction of the latter with V112$_{CXCR4}$ since these two interactions are coupled. Accordingly, V112A leads to only a small change in the IC$_{50}$ values, whereas V112L causes a nearly sixfold increase in the IC$_{50}$ values, indicating that the smaller amino acids (Val or Ala) in this binding pocket are tolerated for the interaction with the sidechain of L1$_{EPI-X4}$, unlike the larger L112 residue. Replacement of D187 by an amino acid with the same charge (D187E) has low or no effect on EPI-X4 binding. Interestingly, the D262A mutation did not lead to a loss of activity. There, we can speculate that the conformational flexibility associated to the introduction of the smaller alanine residue results in larger structural rearrangements allowing new stabilizing interactions between the peptide and the receptor. Furthermore, unlike D187 and D97, D262 seems to play a less prominent role in the interactions with the peptide (Table S2).

Another mutation, L41A, results in a nearly 20-fold increase in IC$_{50}$. In the NTER-IN model, the hydrophobic L41 sidechain is packed with the hydrophobic region of V2$_{EPI-X4}$, suggesting that this interaction is relevant for the formation of the CXCR4/EPI-X4 complex. Therefore, it may be expected that a double mutation involving L41 (V112A + L41A) or a triple mutation (D97E + V112A + L41A) would increase the IC$_{50}$ further. In contrast, these mutations seem to have recovered the loss of L41A activity. In the case of the double mutant, the smaller sidechain of A112 offers flexibility to the interactions involving the L1$_{EPI-X4}$ residue, which in turn allows V2$_{EPI-X4}$ to establish contacts with the V41A sidechain. The flexibility of L1$_{EPI-X4}$ might increase in the triple mutant, where the D97E sidechain is slightly extended, compared to the wild type. These synergistic effects could explain the trends observed in the L41A mutations. Detailed computational studies of the CXCR4 mutants are needed to fully rationalize the effect of such modifications on the interaction networks involved in ligand binding. Nevertheless, the mutagenesis experiments (Table 1) corroborated our computational results indicating that the first seven amino acids of EPI-X4 are essential for binding and that NTER-IN is the favored binding mode of EPI-X4 in CXCR4. Additionally, we also employed NMR spectroscopy to check if there is any change in the structure of EPI-X4 in sodium acetate (to mimic the carboxylate groups of CXCR4 residues). However, the results showed that there is no significant difference in the chemical shift

or the 3D structure of EPI-X4 (Figs. S15–S18), due to the presence of carboxylate ions.

**Analysis of optimized EPI-X4 derivatives WSC02 and JM#21.** Next, we investigated the interactions of the optimized EPI-X4 derivatives WSC02 and JM#21, that showed potent therapeutic effects in mouse models of atopic dermatitis and asthma[16,20]. Both derivatives, WSC02 and JM#21 (also known as EPI-X4 JM#21) are C-terminally truncated analogs of EPI-X4 harboring 12 amino acids each. WSC02 is an optimized derivative with four amino acid substitutions (L1I, Y4W, T5S, and Q10C) and has about 30-fold increased activity compared to its precursor. EPI-X4 JM#21 is an optimized variant of WSC02 with three additional amino acid substitutions (V2L, K6R, and V8L) and is about 35-fold more active than WSC02[20]. First, we performed MD simulations of these peptides to compare their interaction patterns with respect to EPI-X4. Given the structural similarities between the peptides[20], the binding modes of WSC02 and JM#21 were generated by homology modeling using the NTER-IN motif. In both cases, the complexes were subjected to 600 ns MD simulations (three replicas of 200 ns each).

Like in EPI-X4, the RMSD analysis indicated that the structure of CXCR4 is conserved during the simulations (Fig. S6a, c). Although for the peptides the RMSD values indicated structural fluctuations (Figure S6b, d), these are largely focused in the C-terminal region (residues 7–12) of the peptides, which rearranges from the initial conformation, whereas the N-terminal segment (residues 1–6) did not change with respect to the initial pose. This conformational flexibility of the C-terminal region was also observed in the simulations of EPI-X4.

The clustering analysis indicated for both, WSC02/CXCR4 and JM#21/CXCR4, two predominant clusters of structures (populations of 29%,18% and 39%,16%, respectively), which differ only in the orientation of the C-terminal region of the peptide (Fig. S7). We found that, with respect to the parental peptide and WSC02, JM#21 exhibited a somewhat larger amount of conserved hydrogen bonds. According to the model, R6$_{JM\#21}$ interacts with D262$_{CXCR4}$ during 81% of the simulation time (Table S4). WSC02 and JM#21 both form hydrogen bonds with E288$_{CXCR4}$ through their N-terminal amino acid I1 (Table S4 and Fig. 5), which was confirmed experimentally by mutagenesis analysis (Table S5).

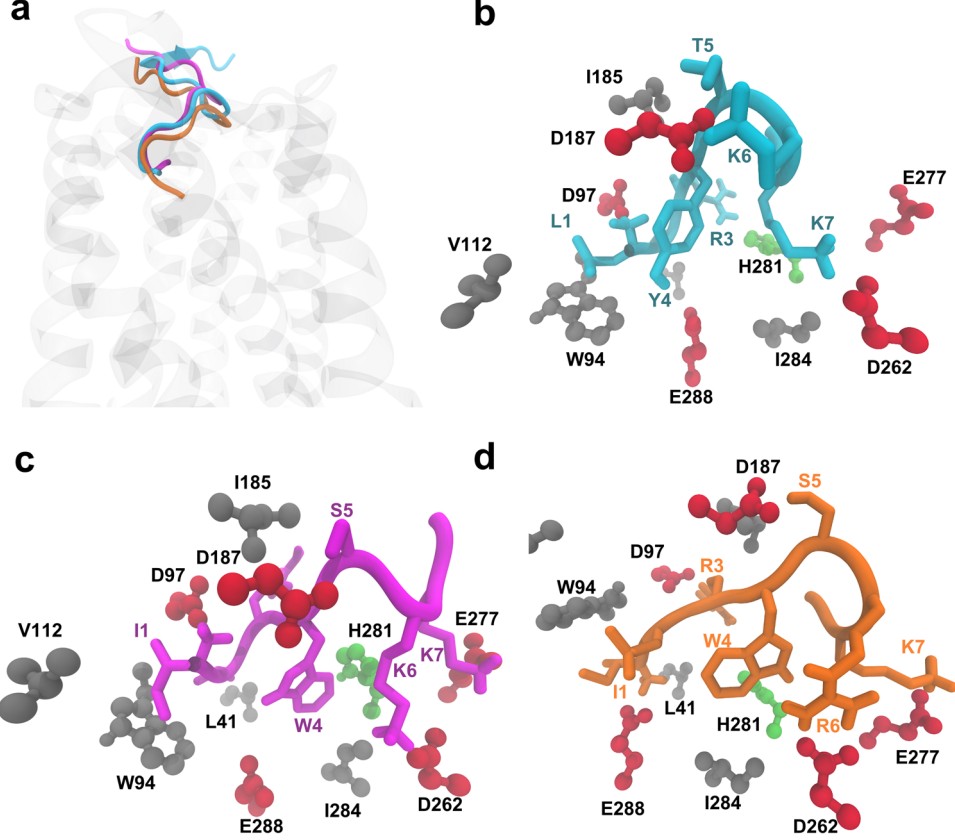

**Fig. 5 Comparison of the binding modes of EPI-X4 (cyan), WSC02 (purple), and JM#21 (orange). a** The position of the three peptides in the binding pocket, **b** EPI-X4, **c** WSC02, and **d** JM#21 (CPK representation of CXCR4 residues by type: acidic residues = red, polar uncharged = green, nonpolar residues = gray).

Substitution of E288 to Ala or Asp nearly abolished binding of WSC02 and JM#21 to CXCR4. A hydrogen bond is also established between $I1_{WSC02}$ and $D97_{CXCR4}$ (Table S4). In our experimental setup, the substitution of D97 strongly decreased the binding of the peptides to CXCR4, similar to the results obtained with EPI-X4. Overall, WSC02 and JM#21 bind to CXCR4 like EPI-X4 through their N-termini.

WSC02 and JM#21 also show similar interaction patterns involving their residues S5, K6/R6, and K7. In the three peptides, the interaction of T5 or S5 with $D187_{CXCR4}$ is conserved. Experimentally, the substitution of D187 by Glu did not have any significant influence on peptide binding. Substitution of D187 to Ala lead to interruption of receptor expression and, thus, could not be tested. The most notable difference among the peptides, WSC02 and JM#21, is the formation of a strong bidentate hydrogen bond between $R6_{JM21}$ and $D262_{CXCR4}$. This could explain the improved binding affinity of JM#21 compared to WSC02 and EPI-X4. The analysis of the interaction energies indicates similar tendencies for WSC02 and JM#21, although the JM#21/CXCR4 complex is more favored than WSC02/CXCR4 by 10 kcal/mol (Fig. S8), displaying an increased contribution of R6 of JM#21 to the binding energy with respect to K6 in WSC02. Similar to the results with EPI-X4, the mutagenesis experiments indicated an improved binding affinity upon introducing the D262A mutation in both cases, WSC02/CXCR4 and JM#21/CXCR4 (Table S5 and Fig. S9). In the absence of secondary interactions reinforcing the binding of K6/R6 to D262 (unlike the case of D97 and D187), this effect could be related to conformational changes in the mutated receptor/peptide complex allowing K6/R6 to establish new salt bridges with other negatively

charged residues within the binding pocket. Thus, the increased receptor binding activity of WSC02 and JM#21 as compared to EPI-X4 appears to be due to the optimal positioning of K6/R6 within the binding pocket of CXCR4. In the case of JM#21, the superior activity can be attributed to the R6 residue.

In comparison with the known antagonists of CXCR4, vMIP-II[11], and IT1t[12], all the three peptides (EPI-X4, WSC02, and JM#21) establish analogous interactions with the binding pocket residues, as shown in Fig. 5a–d. Therefore, it may be speculated that the mechanism of action of EPI-X4 derivatives is similar to that of vMIP-II and IT1t antagonists.

**Rational design of shorter EPI-X4 derivatives**. As discussed above, our results indicate that only the first seven amino acids of EPI-X4 and its improved derivatives WSC02 and JM#21 are involved in receptor binding. Thus, truncations as far as up to position 8 might be even possible without a considerable decrease in affinity. However, an earlier study showed that the truncated derivatives of EPI-X4 (408–415 and 408–414) did not have any significant activity in blocking CXCR4-tropic HIV-1 infection[16]. Based on our computational model of the EPI-X4/CXCR4 interactions, we observed that the shortening of the C-terminal region would bring the carboxylate group of EPI-X4 close to the predominantly negative-charged binding pocket of CXCR4, which might reduce the binding affinity. This trend is in agreement with earlier work[16]. Therefore, to reduce the electrostatic repulsion between the C-terminal group of the peptide and the binding pocket of CXCR4, we neutralized the C-terminal group by amidation.

**Table 2 Length-optimized derivatives of EPI-X4, WSC02, and JM#21.**

| Derivative | Sequence | MW | Length | 12G5-competition (nM) ± SEM | Hill slope[a] |
|---|---|---|---|---|---|
| EPI-X4 (Alb408-423) | LVRYTKKVPQVSTPTL | 1832 | 16 | 2394.0 ± 227.2 | −1.57 ± 0.05 |
| EPI-X4 408–416-NH$_2$ | LVRYTKKVP-NH$_2$ | 1102 | 9 | 1034.3 ± 113.0 | −1.35 ± 0.23 |
| EPI-X4 408–414-NH$_2$ | LVRYTKK-NH$_2$ | 906 | 7 | 954.3 ± 65.3 | −2.25 ± 0.44 |
| WSC02 | IVRWSKKVPCVS | 1401 | 12 | 538.7 ± 150.6 | −2.01 ± 0.07 |
| WSC02 408–416-NH$_2$ | IVRWSKKVP-NH$_2$ | 1111 | 9 | 311.6 ± 119.6 | −2.12 ± 0.73 |
| WSC02 408–414-NH$_2$ | IVRWSKK-NH$_2$ | 915 | 7 | 468.7 ± 164.8 | −5.05 ± 2.63 |
| JM#21 | ILRWSRKLPCVS | 1458 | 12 | 144.8 ± 19.9 | −2.02 ± 0.32 |
| JM#21 408–416-NH$_2$ | ILRWSRKLP-NH$_2$ | 1167 | 9 | 262.4 ± 64.9 | −1.76 ± 0.40 |
| JM#21 408–414-NH$_2$ | ILRWSRK-NH$_2$ | 957 | 7 | 156.8 ± 26.0 | −1.79 ± 0.19 |

[a]Hill slopes were determined in GraphPad Prism using nonlinear regression curve fit. $P > 0.05$ for all hill slopes compared to wt (one-way ANOVA with Dunnett's post hoc test).

Accordingly, based on the structural knowledge of the binding of EPI-X4 and derivatives to CXCR4 and the individual contributions of specific residues to the binding, we designed a series of C-terminally truncated derivatives of EPI-X4, WSC02, and JM#21 and experimentally assessed their activity (Table 2). Derivatives were truncated by seven or nine residues (EPI-X4), or by three or five residues (WSC02 and JM#21), thereby creating analogs that are nine or seven amino acids long, respectively.

First, to reproduce previous results, EPI-X4 with serially truncated C-terminus (but without amidation) was tested again for CXCR4 receptor affinity using the antibody competition assay[33]. Interestingly, C-terminal truncation of EPI-X4 of up to nine amino acids (EPI-X4 408–414) did not lead to a decreased but rather increased binding affinity to CXCR4 confirming our predictions (Fig. S10). Previous results[16] indicated that a C-terminal truncation of no more than four residues was tolerated for HIV-1 inhibition. Here, using the antibody competition assay we could show that also smaller EPI-X4 derivatives have a high affinity for the receptor. In agreement with our modeling results, C-terminal neutralization of truncated EPI-X4 leads to further increased receptor binding (Table 2). As expected, the elimination of K7 (EPI-X4 408-413), almost completely abolished receptor binding, confirming our computational modeling results. Taken together, this suggests that the N-terminal segment of EPI-X4 is highly conserved for the recognition of CXCR4, in agreement with our results indicating that NTER-IN is the most favored binding mode. Indeed, C-terminal truncated and amidated versions of both, EPI-X4 and WSC02, replaced the CXCR4 antibody 12G5 with a slightly increased activity compared to the longer versions. For JM#21, the shorter seven-amino-acid-long analog interacted with the receptor as strongly as the parental 12 amino acids peptide (Fig. S11 and Table 2). Notably, all peptides were fully soluble at concentrations up to 1 mM (Fig. S12).

In the solution structures of EPI-X4[16] and WSC02[20], the N-termini are engaged in contacts within their respective peptide chains. However, both bear an immobilized N-terminus and show structural differences. L1, Y4, and P9 cause an internal ring-like structure ranging from L1 to P9 with a "tail" formed by residues V11–L16 in the case of EPI-X4. In comparison, WSC02 forms a ring-like structure involving the complete peptide chain. This ring is stabilized with a hydrogen bond interaction between the sidechains of S12 and W4 and a salt bridge between the carboxylate group of S12 and the amino group of I1. In contrast, the NMR structure of JM#21[20] reveals a free and flexible N-terminus, which might be more available for receptor binding.

Next, we tested if those length-optimized derivatives are also functionally active CXCR4 antagonists. Downstream signaling of CXCR4 is activated by CXCL12 binding and involves phosphorylation of the signaling proteins Erk and Akt. In the presence of

EPI-X4 JM#21 and its truncated derivatives, CXCL12-mediated activation of both signaling proteins is dose-dependently blocked (Fig. 6a, b and Fig. S13). To see if also chemotaxis can be effectively inhibited, we incubated cancer T lymphoblasts with different concentrations of JM#21 and its truncated analogs and tested for migration towards physiological concentrations of CXCL12 (Fig. 7). As expected, JM#21 inhibited cell migration in a dose-dependent manner, showing 90 ± 6% inhibition at a concentration of 10 μM, in agreement with previous results[20]. Interestingly, the shorter versions of JM#21 (408–416-NH$_2$ and 408–414-NH$_2$) inhibited migration almost completely at the same concentration (96 ± 2% and 99 ± 1%, respectively) and already to 55 ± 6 and 88 ± 2% at a concentration of 1 μM.

Finally, we evaluated the toxicity of EPI-X4 JM#21 408–414-NH$_2$ in zebrafish embryos, a widely used in vivo model for toxicity studies. We exposed embryos for 24 h starting at 24 h post fertilization (hpf), when most organ systems have already developed and are functional. Transparency of the embryos allows for evaluation by light microscopy not only of mortality, but also of sublethal cytotoxicity (necrosis, lysis), developmental toxicity (developmental delay, malformations), or toxicity affecting specific organ systems, in particular cardiotoxicity (heart edema, reduced circulation), and neurotoxicity (reduced touch escape response). A standardized scoring system (Supplementary Table S6) together with the possibility of investigating embryos on a large scale, yields statistically solid and reproducible results and showed that EPI-X4 JM#21 408–414-NH$_2$ is nontoxic even at high concentrations (Fig. S14).

Thus, our novel derivatives encompassing only seven amino acids are more potent in terms of inhibiting CXCR4-downstream signaling and cancer cell migration than the parental peptide, with no toxic effects shown for the shortest JM#21 derivative in the zebrafish toxicity assay.

## Discussion

We here present an experimentally proven structural model of the interaction of endogenous EPI-X4 with its receptor CXCR4, based on atomistic biomolecular simulations. EPI-X4 is the first endogenous peptide antagonist of a chemokine receptor and likely plays a key role in CXCR4 regulation. Our results show that EPI-X4 and its improved derivatives interact with CXCR4 mainly via the first seven amino acid residues, forming interactions with the minor binding pocket of the receptor, in a manner similar to the binding of the viral chemokine antagonist vMIP-II (PDB ID: 4RWS)[11] and the small molecule antagonist IT1t (PDB ID: 3ODU)[12]. These interactions not only occupy the receptor in a way that prevents 12G5 antibody binding to ECL2 but also binding of CXCL12, which explains EPI-X4's CXCR4 antagonizing activity.

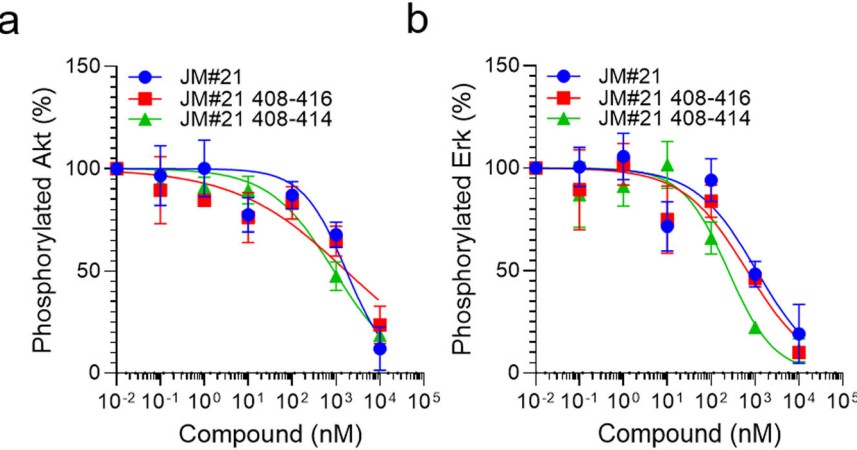

**Fig. 6 Truncated JM#21 variants dose-dependently inhibit CXCL12-mediated signaling.** SupT1 cells were stimulated with 100 ng/ml CXCL12 in the presence of peptides for 2 min. Afterward, the reaction was stopped by adding 2% PFA and shifting the cells to 4 °C. Cells were then permeabilized and subsequently stained with antibodies against pAkt (**a**) and pErk (**b**) for analysis in flow cytometry. Shown are data derived from three individual experiments ± SEM.

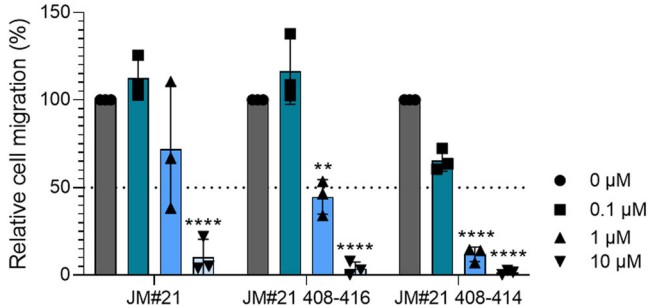

**Fig. 7 Truncated JM#21 variants dose-dependently inhibit CXCL12 induced migration of cancer T cells.** The migration of SupT1 cells towards a 100 ng/ml CXCL12 gradient in a transwell was tested in the presence of indicated concentrations of peptides. After 4 h amounts of migrated cells were determined by CellTiterGlo assay and normalized to values obtained for the PBS control. Shown are values derived from three individual experiments performed in triplicates ± SEM. ****$p < 0.0001$, **$p < 0.01$ (one-way ANOVA with Dunnett's multiple comparison test, compared to PBS control).

Inhibition of CXCR4 is a promising strategy for the treatment of several disorders such as cancers, HIV, and inflammatory diseases. Here, we rationally designed shortened and highly active analogs of EPI-X4 JM#21, a previous lead compound that showed potent therapeutic effects upon topical application in animal models of CXCR4 associated inflammatory diseases, i.e., atopic dermatitis and allergic asthma[20]. The truncated EPI-X4 JM#21 derivatives antagonized cancer cell migration towards CXCL12 even more potently than the precursor peptide. The shortest and most active version (JM#21 408–414-NH$_2$), which exhibited no toxicity in zebrafish assays, encompasses only seven amino acids and has a molecular weight of 957 Da. Interestingly, almost no orally administered drugs and clinical candidates exist with molecular weights exceeding 1000 Da[34]. Thus, our rationally-designed truncated EPI-X4 derivatives should be easier to handle, with less production costs and may pave the way for oral administration of this new class of CXCR4 antagonists.

## Methods

### Computational studies

*CXCR4 model.* The crystal structure of CXCR4 (PDB ID: 3ODU)[12] was used for the modeling studies. However, the N-terminal loop segment (corresponding to

residues 1–26) did not have interpretable electron density and therefore this segment is missing in the crystal structure. Although this segment is not crucial for the interaction of small molecule inhibitors, it might influence the binding of larger molecules such as peptides. For this reason, we modeled the complete structure of the CXCR4 protein by combining the coordinates from the crystal structure with the N-terminal region obtained from NMR studies (PDB ID: 2K04)[32].

*Docking of the EPI-X4 peptide.* The structure of EPI-X4 is available from NMR studies (PDB ID: 2N0X)[16] in solution. From the solution structure ensemble of EPI-X4, three randomly chosen conformations were the starting point for the docking calculations with CXCR4. The HADDOCK web server was used for protein–peptide docking studies[35]. In all cases, residues located at both the minor and major pockets of CXCR4 were chosen as active residues, which serve as the potential site for the interaction with EPI-X4. In the case of EPI-X4, all the residues were considered active. Default docking parameters corresponding to Easy Interface in HADDOCK were used[35]. In addition, one binding mode was built by multi-template homology modeling[36] using the crystal structure of CXCR4 complexed with a viral chemokine antagonist vMIP-II (PDB ID: 4RWS)[11] and the crystal structure 3ODU[12] with the small molecule antagonist IT1t as the second template. The Modeler program (version 9.18) was used for the homology modeling[37].

*Atomistic MD simulations.* Four different binding modes of EPI-X4 were used as starting points for atomistic MD simulations in the lipid bilayer and water. The system consisted of 257 POPC lipids, ~40,000 TIP3P water molecules, 50 mM KCl, and the CXCR4/EPI-X4 complex. The initial system was subjected to energy minimization steps and equilibration MD simulations prior to the production MD runs. For the initial run, harmonic position restraints with a force constant of 10 kcal/mol/Å$^2$ were applied on the protein atoms and the atoms of the lipid head groups. The force constant was gradually reduced to zero in six steps of 200 ps equilibration runs. Production MD simulations were carried out using the equilibrated system. In all cases, periodic boundary conditions were used to eliminate surface effects and the particle mesh Ewald (PME)[38] method was employed for the computation of long-range electrostatics. Short-range Lennard-Jones and electrostatic interactions were cutoff at 12 Å and a switching function was used between 10 and 12 Å to smoothen the interactions at the cutoff distance. Langevin dynamics was employed with the temperature maintained at 300 K and the Langevin piston Nose–Hoover method was used for maintaining the pressure at 1 atm.[39,40] A time step of 2 fs was used for the integration of the equations of motion. All the simulations were replicated three times using different initial random velocities. The CHARMM36 force field[41,42] and the NAMD program (version 2.11)[43] were used.

### Experimental details

*Site-directed CXCR4 mutagenesis and cloning experiments.* The human CXCR4 gene (isoform 1 or b, NCBI Reference Sequence: NP_003458.1, UniProtKB/Swiss-Prot: P61073-1) was amplified by PCR of the pTrip_GFP_CXCR4 vector (kindly provided by Prof. Françoise Bachelerie, Paris, France) by generating the flanking single cutter sites NheI and HindIII. The PCR fragment was ligated in the empty pcDNA3.1(+) vector (Life Technologies GmbH, Darmstadt). Afterward, the IRES-eGFP cassette of the proviral clone pBR_NL4-3_IRES-eGFP[44] was PCR-amplified with EcoRI and NotI single cutter sites and ligated in the multiple cloning site after CXCR4. Site-directed mutagenesis (New England Biolabs, E0554S) was used to

introduce different point mutations in this construct (L41A, D97E, D97S, D97T, D97N, D97Q, V112A, V112L, D187E, D262A, and E277A). Additionally, for some constructs, a second and third round of site-directed mutagenesis was performed to introduce two or three amino acid changes in CXCR4 (D97E plus V112A, D97E plus V112L, L41A plus V112A, and L41A plus D97E plus V112A). As a negative control, a CXCR4 construct harboring two stop codons after the start codon and a mutation introducing a frameshift was cloned (vector only control). All primers are listed in Table S7. All constructs were sequenced to verify their accuracy.

*Peptide synthesis.* Peptides were synthesized on a 0.10 mM scale using standard Fmoc solid-phase peptide synthesis techniques with the microwave synthesizer (liberty blue; CEM). Peptides were purified using reverse phase preparative high-performance liquid chromatography (HPLC). Peptides were lyophilized, mass was verified by liquid chromatography-mass spectroscopy (LCMS), and peptide resolved in PBS before usage.

*Cell culture.* HEK293T cells were cultured in DMEM supplemented with 10% fetal calf serum (FCS), 100 units/ml penicillin, 100 µg/ml streptomycin, and 2 mM L-glutamine (Gibco). SupT1 suspension cells were cultured in RPMI supplemented with 10% FCS, 100 units/ml penicillin, 100 µg/ml streptomycin, 2 mM L-glutamine, and 1 mM HEPES (Gibco).

*Antibody competition assay.* To test for peptide interaction with CXCR4, 50,000 SupT1 cells were seeded in 96-well microtiter plates in PBS supplemented with 1% FCS, the buffer was removed and cells were precooled at 4 °C. Peptides were diluted in precooled PBS before 15 µl were added to the cells together with 15 µl APC-conjugated CXCR4 antibody (clone 12G5; #555976, PD PharmingenTM) at a concentration close to its $EC_{50}$. Cells were incubated at 4 °C for 2 h. Afterward, the unbound antibody was removed and cells analyzed by flow cytometry. The mean fluorescence intensity (MFI) was normalized to the PBS control. Values were normalized to PBS control + antibody. $IC_{50}$ values were determined by nonlinear regression using GraphPad Prism. Antibody competition with CXCR4 mutants 293 T cells were transiently transfected with pcDNA3.1 containing eGFP and CXCR4 wt or CXCR4 harboring selected point mutations. As a control, the vector without functional CXCR4 (vector only) or only transfection reagent (mock) was used. The next day, the medium was changed. Cells were harvested 1 day after, washed, and 50,000 cells were seeded in V-well microtiter plates. Antibody competition was performed as described before[33]. For analysis, after flow cytometry, eGFP expressing cells were gated and further analyzed. CXCR4 expression levels and antibody binding to CXCR4 was determined using the CXCR4 antibody clones 12G5 (binds to the second ECL loop) (#555976, BD PharmingenTM) and 1D9 (binds to the N-terminus) (#551510, BD PharmingenTM) (Supplement Figure S5). For the binding experiment, cells with impaired antibody binding or eGFP expression levels compared to the wild-type control were excluded. After antibody competition, MFIs were normalized to the PBS control + antibody. $IC_{50}$ values and hill slopes were determined by nonlinear regression. ***$p < 0.05$ (one-way ANOVA with Dunnett's post hoc test).

*ERK/AKT signaling assay.* CXCL12 induced ERK and AKT phosphorylation was determined in SupT1 cells. For this, 100,000 cells were seeded per well in a 96-V well plate in 100 µl medium supplemented with 1% FCS. Cells were incubated for 2 h at 37 °C before 5 µl of compounds were added. After 15 min incubation at 37 °C cells were stimulated by adding 5 µl CXCL12 diluted in PBS to reach a final concentration of 100 ng/ml. Cells were further incubated for 2 min before the reaction was stopped by adding 20 µl of 10% PFA. Cells were fixed for 15 min at 4 °C before PFA was removed and cell permeabilized by adding 100 µl ice-cold methanol. After 15 min at 4 °C, the methanol was removed, cells were washed, and 30 µl primary antibody was added (phospho-p44/42 MAPK (Erk1) (Tyr204)/(Erk2) (Tyr187) (D1H6G) mouse mAb #5726; phospho-Akt (Ser473) (193H12) rabbit mAb #4058 Cell Signaling) for 1 h at 4 °C. After the antibody was removed and cells were washed secondary antibody was added for 30 min. Cells were washed afterward and subsequently analyzed by flow cytometry.

*Migration of cancer T lymphoma cells.* Migration assays towards a 100 ng/ml CXCL12 gradient (#300-28 A, Peprotech) were performed using 96-well transwell assay plates (Corning Incorporated, Kennebunk, ME, USA) with 5-µm pore polycarbonate filters. First, 50 µl (0.75 × 105) SupT1 cells resuspended in assay buffer (RPMI supplemented with 0.1% BSA) were seeded into the upper chamber in the presence or absence of compounds and allowed to settle down for around 15 min. In the meantime, 200 µl assay buffer supplemented with or without 100 ng/ml CXCL12 as well as compounds were filled into a 96-well-V plate. Cells were allowed to migrate towards CXCL12 by putting the upper chamber onto the 96-well-V plates. After a migration time of 4 h at 37 °C (5% CO2), the lower compartments were analyzed for cell content by Cell-Titer-Glo® assay (Promega, Madison, WI, USA). Percentages of migrated cells were calculated as described before[45] and normalized to the CXCL12-only control.

*Toxicity assays in zebrafish.* Wild-type zebrafish embryos were dechorionated at 24 hpf using digestion with 1 mg/ml pronase (Sigma) in E3 medium (83 µM NaCl,

2.8 µM KCl, 5.5 µM CaCl2, and 5.5 µM MgSO4). Embryos were exposed for 24 h, in groups of 3, to 100 µl of E3 containing JM#21 408–414-NH2 at 3, 30, and 300 µM. Each concentration was tested in two independent assays, each of which was performed on 10 × 3 embryos. The peptide solvent (PBS), diluted in E3, was used as a negative control at the same amount as introduced by the highest peptide concentration. The pleurocidin antimicrobial peptide NRC-03 (GRRKRKWLRRIGKGVKIIGGAALDHL-NH2) was used as a positive control for cytotoxicity at a concentration of 6 µM as described[46]. Abamectin at a concentration of 3.125 µM was used as a positive control for neurotoxicity[47]. At 48 hpf (after 24 h of incubation) embryos were scored in a stereomicroscope for signs of cytotoxicity (lysis and/or necrosis), developmental toxicity (delay and/or malformations), or cardiotoxicity (heart edema and/or reduced or absent circulation). Each embryo was also touched with a needle, and reduced or absent touch responses (escape movements) were evaluated as signs of neurotoxicity if and only if no signs of cytotoxicity were present in the same embryo. Embryos were categorized within each of these toxicity categories into several classes of severity according to the criteria listed in Supplementary Table S7. The Chi-Square test was used to calculate whether the distribution of embryos into toxicity classes differed significantly between the PBS negative control and the test substances.

*Peptide solubility by BCA assay.* To determine the solubility of peptides in $H_2O$, PBS, or in the presence of 1% Tween20 in PBS, peptides were serially diluted in the different solvents and incubated at 4 °C for 24 h. Afterward, mixtures were centrifuged at 20,000x$g$ for 5 min and supernatants were analyzed by BCA assay (Pierce Rapid Gold BCA Protein Assay Kit, Thermo Fisher) according to the manufacturer's instructions. For comparison, the peptide was freshly dissolved, serially diluted, and analyzed in parallel.

*Statistics and reproducibility.* Statistics were performed using the PROC ANOVA procedure in SAS, version 9.4 (www.sas.com). All data performed in cell culture were obtained in at least three independent experiments. For CXCL12-dependent assays (signaling or migration) assays were performed in triplicates.

**Reporting Summary.** Further information on research design is available in the Nature Research Reporting Summary linked to this article.

## Data availability

All raw data and histograms that support the findings of this study are available from the corresponding authors upon reasonable request. Structural data from the simulations is available as supplementary material. Atomic coordinates of the binding modes of the EPI-X4–CXCR4 complex are available in Supplementary Data 1. Raw data derived from the flow cytometry analysis in Figs. 4, 6, and 7 is available in Supplementary Data 2.

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

## Acknowledgements

This work was supported by the Deutsche Forschungsgemeinschaft (DFG, German Research Foundation) under the collaborative research center CRC 1279 (project A06). E.S.-G. was also supported by Germany´s Excellence Strategy – EXC 2033 – 390677874 – RESOLV and the collaborative research center CRC 1093 "Supramolecular Chemistry on Proteins", both funded by the DFG. E.S.-G. acknowledges the support of the Boehringer Ingelheim Foundation (Plus-3 Program) and the computational time provided by the Computing and Data Facility of the Max Planck Society and the supercomputer magnitUDE of the University of Duisburg-Essen. P.S. acknowledges the Department of Science and Technology in India for the support received through DST-SERB SRG/2019/002156. F.K. acknowledges funding by the ERC PoC grant "EPI-X4 Health". J.M. further acknowledges funding by the Baden-Württemberg Stiftung, the European Research Council, and the DFG grant MU 3315/11-1. M.H. is part of the International Graduate School in Molecular Medicine Ulm.

## Author contributions

P.S. performed the simulations, analyzed the data, and wrote the manuscript together with M.H. and E.S.-G. M.H. performed the experiments, analyzed the experimental data, and wrote the manuscript together with P.S. and E.S.-G. C.S. did the cloning experiments, A.G. performed the migration essays, G.K. contributed the NMR studies, N.P and L.S. did the peptide synthesis and purity control. M.R. and G.W. performed the zebrafish toxicity studies. B.M. performed statistical analyses. F.K. supervised experimental work. J.M. coordinated and supervised the experimental work and designed the project together with E.S.-G. E.S.-G. designed the project, coordinated and supervised the work, analyzed the data, and wrote the manuscript with P.S and M.H.

## Funding

## Competing interests

Authors P.S., M.H., L.S., F.K., J.M. and E.S.-G. are inventors of granted or pending patents claiming to use EPI-X4 and optimized derivatives for the therapy of CXCR4-linked diseases.
