## [Transparent Peer Review File · Communications Biology]

Reviewers' comments:

Reviewer #1 (Remarks to the Author):

The manuscript by Sokkar et al. describes an in silico and in vitro analysis of the interaction between the chemokine receptor CXCR4 and several variants of EPI-X4, a fragment of serum albumin that has previously been shown to be a weak endogenous CXCR4 antagonist. The work capitalizes on earlier SAR studies of EPI-X4 by the same group, as well as a computational modeling effort that has elucidated a likely interaction geometry. In this work, molecular dynamics simulations are conducted to explore several alternative hypotheses for EPI-X4 binding in the CXCR4 TM domain pocket; following these simulations, the authors narrowed down on the "N-terminus-in" hypothesis as the most likely as it produced the most stable system behaviors, a larger number of predicted favorable interactions than other hypotheses, and was more consistent with peptide SAR and receptor mutagenesis. Next, under the "N-terminus-in" assumption, the authors discovered that shortening the C-terminus of EPI-X4 and its variants while simultaneously keeping it electrically neutral via amidation can produce equally or more potent analogs with smaller molecular weight: an important step towards making EPI-X4 derivatives into therapeutic candidates. The most potent of the designed analogs had IC50 of ~156 nM in the competition binding assay with 12G5, an antibody targeting the extracellular loop 2 of the receptor. They also inhibited pErk and pAkt downstream of CXCL12-stimulated CXCR4 with IC50 ~ 100-1000 uM, and inhibited transwell migration of SupT1 cells relatively weakly but in a dose-dependent manner. Finally, the best analog was found non-toxic to zebrafish embryos when applied 24 hours post fertilization (after most of the organ systems have formed) for 24 hours.

This is a solid multidisciplinary study and the results support the conclusions reasonably well. One concern for this reviewer is the non-canonical shape of the 12G5 competition binding curves, as in most cases, they do not seem to plateau at the highest concentrations of the peptide analogs tested, and appear to have high Hill slopes, potentially indicating aggregation issues or nonspecific binding. To this effect, it is important that at least some of the experiments are conducted in settings preventing colloidal aggregation, e.g. with the use of detergents and following peptide precipitation by spinning. Hill slopes for all complete binding curves need to be presented numerically along with the calculated IC50s. Statistical significance of changes in the fitted curve parameters needs to be assessed using one of the established methods (e.g. F test).

One possible explanation for the strangely shaped competition binding curves is a suboptimal choice of the probe. The 12G5 Ab binds "externally" to the ECL2, without penetrating the binding pocket, which makes it possible that ternary complexes (receptor, antibody, peptide analog) form. To address this possibility, competition binding experiments with the chemokine itself (radiolabeled, fluorescently labeled, or tagged and detected by a fluorescent anti-tag antibody) would help; such experiments would also provide a direct way to correlate the analog's affinity with their ability to antagonize CXCL12-triggered functional responses.

Additional changes that need to be made to improve the manuscript clarity, transparency and compliance with the rigor/reproducibility requirements:

- * Flow cytometry histograms (or 2D histograms for two-color experiments) need to be presented in the Supplementary Materials.
- * Biological replicates need to be shown as individual points on all bar graphs.
- * Model coordinates and sparsely sampled trajectories need to be made public either as supplementary materials or by depositing them to one of the existing servers for MD simulations (e.g. GPCRMD).

Other minor comments:

- * p.31: "phosphor-Akt" -> "phospho-Akt"
- * p.32: "5 µm polycarbonate filters" -> "5 µm pore polycarbonate filters"

Reviewer #2 (Remarks to the Author):

Combining molecular modeling and a series of in vitro bioassays, in this manuscript Sokkar et al. provided an understanding of how EPI-X4, an endogenous peptide antagonist, interacts with its receptor CXCR4. They employed MD simulation to build a structural model of EPI-X4 bound to CXCR4 to derive detailed information about peptide-receptor interaction. They verified their computational models through mutagenesis experiments. Furthermore, based on the model, they designed and synthesized a series of derivatives and carried out comparative binding and functional assays to test their inhibition of CXCR4 function to obtain results consistent with the computer prediction.

A couple of minor issues that need revision :

1. What is the difference between Figures 5 (A, C, E) and 7? This should be illustrated clearly in the legend.
2. In Figure 5, graph B, D, F may not visually show the drastically decrease of binding in some mutations. It is suggested to provide a table that lists the IC50 values and folds of change as compared with WT of all the mutants, and discuss them in the text also.

Reviewers' comments (*in italics*) and authors answers (in dark blue):

Reviewer #1 (Remarks to the Author):

The manuscript by Sokkar et al. describes an in silico and in vitro analysis of the interaction between the chemokine receptor CXCR4 and several variants of EPI-X4, a fragment of serum albumin that has previously been shown to be a weak endogenous CXCR4 antagonist. The work capitalizes on earlier SAR studies of EPI-X4 by the same group, as well as a computational modeling effort that has elucidated a likely interaction geometry. In this work, molecular dynamics simulations are conducted to explore several alternative hypotheses for EPI-X4 binding in the CXCR4 TM domain pocket; following these simulations, the authors narrowed down on the “N-terminus-in” hypothesis as the most likely as it produced the most stable system behaviors, a larger number of predicted favorable interactions than other hypotheses, and was more consistent with peptide SAR and receptor mutagenesis. Next, under the “N-terminus-in” assumption, the authors discovered that shortening the C-terminus of EPI-X4 and its variants while simultaneously keeping it electrically neutral via amidation can produce equally or more potent analogs with smaller molecular weight: an important step towards making EPI-X4 derivatives into therapeutic candidates. The most potent of the designed analogs had IC50 of ~156 nM in the competition binding assay with 12G5, an antibody targeting the extracellular loop 2 of the receptor. They also inhibited pErk and pAkt downstream of CXCL12-stimulated CXCR4 with IC50 ~ 100-1000 uM, and inhibited transwell migration of SupT1 cells relatively weakly but in a dose-dependent manner. Finally, the best analog was found non-toxic to zebrafish embryos when applied 24 hours post fertilization (after most of the organ systems have formed) for 24 hours. This is a solid multidisciplinary study and the results support the conclusions reasonably well.

Reply: We thank this reviewer for appreciating the value and quality of our work.

-One concern for this reviewer is the non-canonical shape of the 12G5 competition binding curves, as in most cases, they do not seem to plateau at the highest concentrations of the peptide analogs tested, and appear to have high Hill slopes, potentially indicating aggregation issues or nonspecific binding. To this effect, it is important that at least some of the experiments are conducted in settings preventing colloidal aggregation, e.g. with the use of detergents and following peptide precipitation by spinning.

Reply: We tested solubility of all relevant peptides in H₂O, PBS and in the presence of a detergent (1 % Tween20). For this, we serially diluted the peptides starting at 1 mM in the respective diluents. After 24 h incubation at 4°C, substances were centrifuged with high speed and the upper fraction (supernatant) used to determine remaining peptide concentrations by BCA assay using a freshly diluted peptide as reference. The data are discussed in lines 422-423 and are presented as new Supplementary Figure 10. No difference between centrifuged and freshly dissolved peptides could be detected. Curves are very linear even at very high concentrations, indicating no aggregation issues at those concentrations. Notably, highest peptide concentration used in the assays is 100 μM. As detergents cannot be used in our assay (which includes living cells), competition assays with peptides dissolved in detergents could not be performed under these conditions.

-Hill slopes for all complete binding curves need to be presented numerically along with the calculated IC₅₀s. Statistical significance of changes in the fitted curve parameters needs to be assessed using one of the established methods (e.g. F test).

Reply: We re-analyzed each binding curve and added all Hill slopes for EPI-X4 binding experiments to Table 2. All IC₅₀ values and Hill slopes were then statistically compared to the values obtained with the wild type receptor using one-way ANOVA (with support of a statistician). For Hill slopes no significant difference was observed if compared to the wild types or among the mutants. Statistics for IC₅₀ values are presented in the new Figure 5.

In addition, the same was done for values obtained with the optimized and truncated peptides (Table 3). Here, no significant difference was detected for all Hill slopes compared to the respective parental peptide. Notably, Figure 7 was updated and now shows antibody competition curves obtained for the optimized peptides.

-One possible explanation for the strangely shaped competition binding curves is a suboptimal choice of the probe. The 12G5 Ab binds “externally” to the ECL2, without penetrating the binding pocket, which makes it possible that ternary complexes (receptor, antibody, peptide analog) form. To address this possibility, competition binding experiments with the chemokine itself (radiolabeled, fluorescently labeled, or tagged and detected by a fluorescent anti-tag antibody) would help; such experiments would also provide a direct way to correlate the analog’s affinity with their ability to antagonize CXCL12-triggered functional responses.

Reply: Competition experiments with the chemokine itself are a very good suggestion. However, CXCL12 binding to CXCR4 follows a 2-step binding mode and results in receptor internalization which may complicate data interpretation; in addition, CXCL12 coupling to fluorophores may alter receptor binding and the generation and purification of such tagged chemokines is time consuming, laborious and expensive. For these reasons, we developed the 12G5 antibody competition assay as fast and reliable assay to determine binding affinities of various CXCR4 ligands in living cells. The assay has been published (Harms et al., Sci Rep 2020, DOI: <https://doi.org/10.1038/s41598-020-73012-4>) and allows a convenient and cheap determination of binding affinities of various CXCR4 antagonists in living cells within just 3 h. Moreover, the assay can be performed in the presence of high concentrations of physiologically relevant body fluids, and thus is a useful readout to evaluate stability (i.e. half-life) of CXCR4 ligands in serum/plasma. This 12G5 antibody-competition assay allows a robust and convenient determination and calculation of pharmacological parameters of CXCR4 receptor-drug interaction and has also been applied to develop optimized EPI-X4 derivatives for therapy of atopic dermatitis and asthma (Harms et al., Acta Pharmaceutica Sinica B 2020, DOI: <https://doi.org/10.1016/j.apsb.2020.12.005>). It is important to mention, that most mutations in the binding pocket of CXCR4 do not affect antibody binding. In addition, we showed that the antibody can be fully replaced by the peptides used in the study under normal conditions (Harms et al., Sci Rep 2020, and Harms et al., Acta Pharmaceutica Sinica B 2020). The reason for the rather high IC₅₀ values and sometimes incomplete competition curves in the present paper is most probably due to the presence of high amounts of CXCR4 receptor on transfected cells. If CXCR4 concentration is way above the K_d of the probe under chosen conditions (~ 151.1 pM, Harms et al., Sci Rep 2020), ligand depletion takes place and might result in incomplete binding curves. However, this issue should not disturb relative binding affinities (relative IC₅₀ values) and comparison of mutant to wild type receptor binding, as transfection rates and antibody staining were comparable among receptor variants. Assays performed with SupT1 cells (Figure 7) were

designed with CXCR4 concentrations < 10% Kd (12G5), so ligand depletion is neglectable (Harms et al. Sci Rep 2020). Also, curves obtained under those conditions are complete.

Thus, our 12G5 antibody competition assay is validated and allows to determine binding of CXCR4 ligands to CXCR4. Instead of using a tagged version of CXCL12 for competition binding experiments, we performed functional CXCL12-dependent signaling studies using the most relevant optimized derivatives (JM#21 variants), since inhibition of CXCL12 induced signaling is more relevant than competition with receptor binding. We show dose-dependent inhibition of CXCL12-induced phosphorylation of Akt and Erk signaling and inhibition of CXCL12-directed cell migration by EPI-X4 analogs. Those experiments confirm data obtained by 12G5-antibody competition for JM#21 variants (Figure 8 and 9). A more detailed pharmacodynamic analysis is beyond the scope of the present study.

*-Additional changes that need to be made to improve the manuscript clarity, transparency and compliance with the rigor/reproducibility requirements: * Flow cytometry histograms (or 2D histograms for two-color experiments) need to be presented in the Supplementary Materials.*

Reply: Exemplary flow cytometry histograms and respective gating strategies are now provided as new supplementary figures 4 and 11. All primary data are available upon request as stated in lines 637-640 of the manuscript.

- Biological replicates need to be shown as individual points on all bar graphs.*

Reply: We now show individual data points in all bar graphs (see new Figs. 5, 9 and S4)

- Model coordinates and sparsely sampled trajectories need to be made public either as supplementary materials or by depositing them to one of the existing servers for MD simulations (e.g. GPCRMD).*

Reply: The coordinates of the structures from the sampled trajectories reported in this work are now available as supplementary material.

-Other minor comments:n p.31: “phosphor-Akt” -> “phospho-Akt”
* p.32: “5 μm polycarbonate filters” -> “5 μm pore polycarbonate filters”*

Reply: We corrected the typos.

Reviewer #2 (Remarks to the Author):

Combining molecular modeling and a series of in vitro bioassays, in this manuscript Sokkar et al. provided an understanding of how EPI-X4, an endogenous peptide antagonist, interacts with its receptor CXCR4. They employed MD simulation to build a structural model of EPI-X4 bound to CXCR4 to derive detailed information about peptide-receptor interaction. They verified their computational models through mutagenesis experiments. Furthermore, based on the model, they designed and synthesized a series of derivatives and carried out comparative binding and functional assays to test their inhibition of CXCR4 function to obtain results consistent with the computer prediction.

A couple of minor issues that need revision :

1. What is the difference between Figures 5 (A, C, E) and 7? This should be illustrated clearly in the legend.

Reply: We thank the reviewer for noting this issue, as in the previous version we included Figure S7 (with extended information with respect to the Figures mentioned by the reviewer) instead of the correct Figure 7. The correct Figure 7 is now included, which shows antibody competition curves for the optimized peptides (see also Table 3).

2. In Figure 5, graph B, D, F may not visually show the drastically decrease of binding in some mutations. It is suggested to provide a table that lists the IC50 values and folds of change as compared with WT of all the mutants, and discuss them in the text also.

Reply: All IC50 values and fold changes are now provided in Table 2 (see above) and Supplementary Table 4.

REVIEWERS' COMMENTS:

Reviewer #1 (Remarks to the Author):

This is a revised version of the manuscript by Sokkar et al. describing an in silico and in vitro analysis of the interaction between the chemokine receptor CXCR4 and selected variants of EPI-X4, a fragment of serum albumin that has previously been shown to be a weak endogenous CXCR4 antagonist. In the revision, the authors adequately addressed comments from this reviewer and included peptide solubility data, flow cytometry scatter plots, individual replicates on graphs, and the analysis of statistical significance. The only suggestion that was not addressed as a competition binding experiment between EPI-X4 and a fluorescent analog of CXCL12 which would be a better probe than the currently used 12G5 because CXCL12 binds to the same binding pocket of CXCR4 as the peptides in question. The authors motivated their decision to not conduct such experiments by (i) CXCR4 internalization upon CXCL12 binding and (ii) the possibility of CXCL12 tagging altering its binding properties. However, the use of fluorescently and luminescently tagged CXCL12 for studies of CXCR4 is actually very well established and the authors should look into this assay as a complement to their currently used 12G5 competition assay (since 12G5 is not even expected to directly compete with the peptides). Some relevant references include:

* Luker 2009 <https://www.ncbi.nlm.nih.gov/pmc/articles/PMC4418468/>

* Kawamura 2014 <https://pubmed.ncbi.nlm.nih.gov/24489642/>

* Schoofs 2018 <https://www.ncbi.nlm.nih.gov/pmc/articles/PMC5931669/>

* Gustavsson 2019 <https://stke.sciencemag.org/content/12/598/eaaw3657?rss=1>

This said, I believe that the presented data in this study is now sufficient to support the conclusions and therefore I am happy to recommend the publication of the article in its present form.